# The A-B transition in superfluid helium-3 under confinement in a thin slab geometry

N. Zhelev[1], T.S. Abhilash[1], E.N. Smith[1], R.G. Bennett[1], X. Rojas[2], L. Levitin[2], J. Saunders[2] & J.M. Parpia[1]

The influence of confinement on the phases of superfluid helium-3 is studied using the torsional pendulum method. We focus on the transition between the A and B phases, where the A phase is stabilized by confinement and a spatially modulated stripe phase is predicted at the A–B phase boundary. Here we discuss results from superfluid helium-3 contained in a single 1.08-µm-thick nanofluidic cavity incorporated into a high-precision torsion pendulum, and map the phase diagram between 0.1 and 5.6 bar. We observe only small supercooling of the A phase, in comparison to bulk or when confined in aerogel, with evidence for a non-monotonic pressure dependence. This suggests that an intrinsic B-phase nucleation mechanism operates under confinement. Both the phase diagram and the relative superfluid fraction of the A and B phases, show that strong coupling is present at all pressures, with implications for the stability of the stripe phase.

[1] Department of Physics, Cornell University, Ithaca, New York 14853, USA. [2] Department of Physics, Royal Holloway University of London, Egham, Surrey TW20 0EX, UK. Correspondence and requests for materials should be addressed to J.M.P. (email: jmp9@cornell.edu).

Superfluid [3]He is one of the richest condensed matter systems. Its impact extends to fields as diverse as unconventional superconductivity[1–5], cosmology[6–11] and turbulence[12]. The [3]He spin-triplet p-wave superfluid order parameter is described by a $3 \times 3$ matrix encoding the orientation of the spin and orbital angular momentum of the Cooper pairs over the Fermi surface.

In zero magnetic field two superfluid phases, A and B, emerge to break the rotational symmetry of the normal state in different ways[13,14]. These superfluids belong to important classes of topological quantum matter and serve as model systems for topological superconductivity[15–17]. The B phase of superfluid [3]He is a time-reversal invariant odd-parity condensate of p-wave pairs, with all three components of the spin and orbital triplet states present, and isotropic energy gap. The A phase is a chiral superfluid, in which only the equal spin pairs form with a common orbital angular momentum $\ell$ vector. In the weak coupling approximation, superfluid [3]He is described in terms of p-wave Bardeen-Cooper-Schrieffer (BCS) theory. In practice, the onset of superfluidity modifies the pairing interaction, for example pairing mediated by the exchange of spin-fluctuations. These strong-coupling effects stabilize the bulk A phase at high pressures[18], but the B phase dominates the bulk phase diagram, being the favoured phase at $T = 0$ at all pressures.

Near a wall, gap-distortions arise from de-pairing due to surface scattering; the A phase orients the $\ell$ vector normal to the surface, minimizing gap suppression[19,20]. This weaker gap suppression favours the A phase and is responsible for the profound influence of confinement. The relevant length scale over which gap distortion occurs is set by the pressure-dependent Cooper pair diameter, $\xi_0 = \hbar v_F / 2\pi k_B T_c$, where $v_F$ is the Fermi velocity, $T_c$ is the bulk superfluid transition temperature, $\hbar$ and $k_B$ are Planck's and Boltzmann's constants. When two surfaces are separated by a comparable distance $D$, $D/\xi_0 \cong 10–20$, the distortion of the order parameter at the surface influences the entire sample, particularly approaching $T_c$, promoting the A phase over the B phase[21].

A further consequence of moderate confinement is the prediction of a spatially modulated or stripe phase, intervening between the A and B phases[21,22], that breaks translational symmetry in the plane of the slab, and is composed of alternating regions of degenerate B phase domains of different orientations. It is an analogue of the Fulde–Ferrel–Larkin–Ovchinnikov phase[23,24], long sought after in superconductors[25,26] and fermionic ultra-cold atom systems[27–29]. In [3]He, the driving mechanism for the spatially modulated phase is the negative surface energy of domain walls under confinement, that allow domains of degenerate B-phase quantum vacua to spontaneously appear. The existence, location and stability of the putative stripe phase have been shown to depend crucially on details of the strong coupling parameters[22].

Until recently investigations of [3]He under regular confinement were limited to arrays of plates and capillaries[30–32], or studies of saturated films[33–35]. Our approach in the study of topological superfluidity is to confine the [3]He in precisely engineered nanofabricated geometries, such as a regular well characterized cavity geometry with height of order the Cooper pair diameter, $\xi_0$ (ref. 36). In a previous experiment, [3]He was confined to a single 680-nm-tall cavity, and nuclear magnetic resonance (NMR) was used to map the phase diagram. The B phase was found to be completely excluded at low pressure and the A phase was observed between the normal state and the B-phase above 3 bar (refs 36,37).

In the following, we report on a study, using a torsion pendulum, to measure the superfluid density under somewhat weaker confinement. The cavity height of 1.08 μm was chosen in the light of previous work[36,37], such that an A–B transition was expected to occur at all pressures. We show the profound

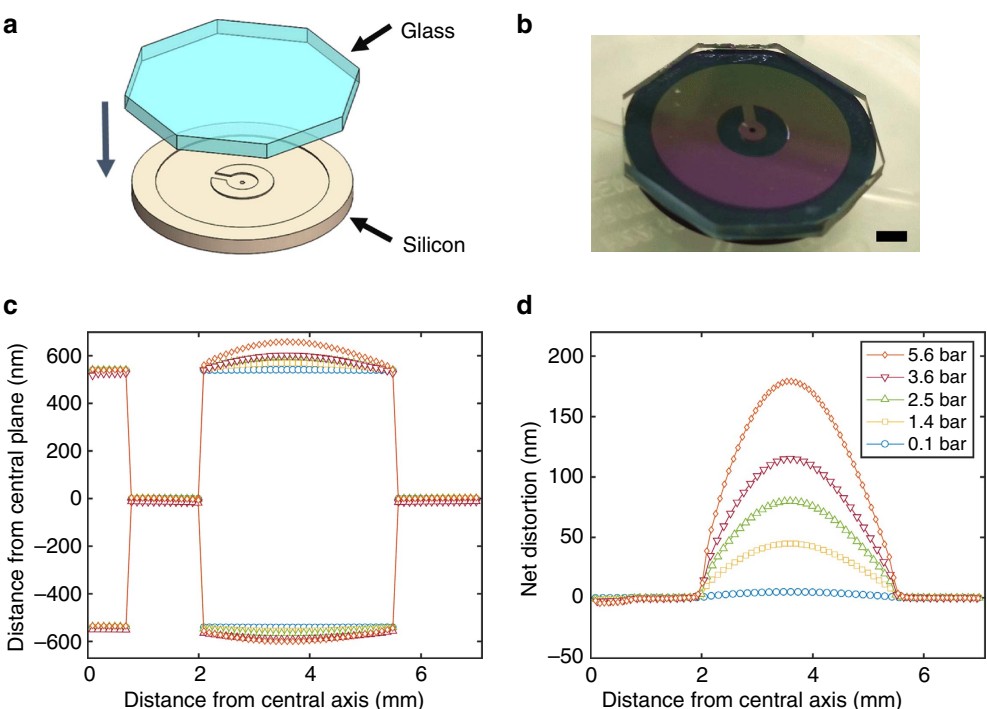

**Figure 1 | Torsion pendulum head.** (**a**) Schematic representation of the glass-silicon head. The 1 mm thick × 14 mm diameter silicon was patterned with a 1.08 μm tall × 11 mm outer diameter/4 mm inner diameter cavity before the octagonal glass lid was anodically bonded to it. The annular cavity is connected to the central fill line through a 1.25 mm long × 0.6 mm wide radial channel that opens into a 1.5 mm diameter central hub. (**b**) The bonded cavity before mounting on the torsion pendulum. Scale bar, 2 mm. (**c**) The cross-section of the cavity under pressure. (**d**) The calculated net bowing of the cavity at the five experimental pressures investigated. (**c,d**) Calculations were done using finite element methods.

influence of confinement on both the A–B phase boundary in zero magnetic field, and the nucleation of B-phase from the A phase. We observe an A to B transition on cooling at all studied pressures in zero magnetic field, as anticipated. However, we make the striking observation that the supercooling of the first-order A–B transition is very small, far less than the considerable supercooling observed in bulk[38], or in anisotropic aerogel[39,40]. This occurs despite a cavity geometry that should isolate the confinement-stabilized A phase from bulk B-phase. The efficient nucleation and non-monotonic supercooling are suggestive of the presence of an intrinsic B-phase nucleation mechanism under confinement. We discuss the insights this brings to the mystery of B-phase nucleation. This is of interest not only as a transition between two different topological phases but also between two quantum vacua of different symmetry, with cosmological analogues to symmetry breaking phase transitions in the early universe responsible for its large scale structure and the dominance of matter over anti-matter[6–11,38,41–47]. Furthermore, from details of the A–B phase boundary under confinement, and superfluid fractions at the transition, we show that strong-coupling effects persist to the lowest pressure. This has possible implications for the stripe phase, not observed directly in our experiment.

## Results

**Experiment details.** The $^3$He was confined to a 1.08-μm-deep cavity micromachined in 1-mm-thick silicon, capped with 1-mm-thick sodium-doped glass, anodically bonded[48] to the silicon. The cavity is shown in Fig. 1a,b and construction details are provided elsewhere[49] and in Supplementary Figs 1 and 2. Under pressure, the cavity distorts by 180 nm at 5.6 bar as found by finite element modelling and depicted in Fig. 1c,d and Supplementary Fig. 3 (see also Methods). The bowing of the cell plays an important role in our study of B-phase nucleation. For the measurements described here, the surfaces were coated with a 30 μmole m$^{-2}$ coverage of $^4$He to eliminate solid $^3$He from the surface[50] and allow direct comparison with the earlier NMR experiment, with diffuse scattering[36,37].

**Measurables.** We measure the resonant frequency, $f$, and the quality factor, $Q$, of the torsion pendulum as a function of temperature. The $^3$He in the cavity is fully coupled to the torsion pendulum above $T_c$. The superfluid fraction is determined from the increase in resonant frequency (after subtracting the temperature-dependent background of the empty oscillator), arising from the decoupling of the superfluid below $T_c$. The dissipation $(Q^{-1})$ was calculated by relating the observed amplitude at resonance to the drive voltage (after calibration at a fixed drive). Temperatures were measured using a melting curve thermometer[51,52] and then related to the frequency shift and dissipation of a quartz fork, immersed in the same heat exchanger as the $^3$He in the torsion pendulum[53]. Below we express temperatures in units of the bulk $T_c$, measured *in situ* by the fork. $T_c = 0.9$–1.6 mK at $P = 0.1$–5.6 bar. The suppression of $T_c$ in the slab due to confinement was less than $0.01 T_c$, and is not discussed further.

**Superfluid fraction.** The superfluid fractions measured while cooling (blue circles) and warming (red open triangles) for five pressures are plotted in Fig. 2. The data highlighted by circles contains the signature of the AB transition shown in detail in Fig. 3. The superfluid fraction (dashed line) of the bulk B phase at each pressure[54] is essentially indistinguishable from that measured under confinement.

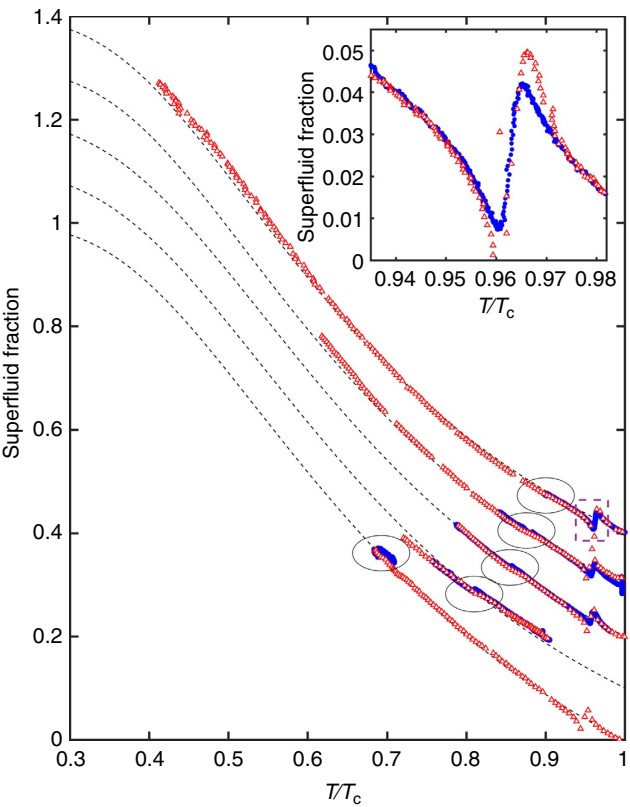

**Figure 2 | Temperature dependence of the superfluid fraction.** Superfluid fraction (each offset by 0.1 from adjacent results for clarity) measured at 0.1 (bottom), 1.4, 2.5, 3.6 and 5.6 bar (top) while warming (open red triangles) and cooling (filled blue circles) as a function of temperature measured in units of bulk superfluid transition temperature $T_c$. Encircled are the locations of the A–B and B–A transitions, shown in detail in Fig. 3. Dashed lines show the superfluid density of the bulk B phase[54]. The anti-symmetric signature just below $T_c$ arises from a mode crossing of a Helmholtz resonance with the torsional mode (marked with a dashed box and shown in inset). The crossing is narrower after warming from the B phase into the A phase than after cooling from the normal fluid into the A phase.

**Relative confinement.** Following Levitin et al.[36], we adopt the temperature-dependent coherence length $\xi_\Delta(T) = \hbar v_F / (10^{1/2} \Delta_B(T))$, where $\Delta_B(T)$ is the bulk superfluid B phase gap parameter. $\xi_\Delta(0) = 1.13\xi_0$, and $\xi_\Delta(T)$ tends to the Ginzburg–Landau (GL) result as $T \to T_c$. The relative confinement is then expressed as $D/\xi_\Delta(T)$, where $D$ is the confining slab's height. The predicted equilibrium AB transition temperature, $T_{AB}$, for slabs of different thickness is given by a universal value of $D/\xi_\Delta(T_{AB})$, which increases with pressure due to strong coupling[36,55–57]. As discussed, the nominal cavity height 1,080 nm, corresponding to $D/\xi_\Delta(T=0, P=0) = 12$, was chosen such that an A–B transition was expected to occur at all pressures.

**Transition between A and B phases.** The region near the first-order AB transition is the focus of this paper. We summarize the results here. The superfluid fraction near the AB transition is shown in Fig. 3 and the values for various quantities determined by our fitting procedure described in Supplementary Note 1 are shown in Fig. 4 and in Table 1. The influence of confinement on the equilibrium transition temperature, $T_{BA}$ observed on warming is listed in Table 1. The transition is broadened by pressure-induced bowing of the cavity (Table 1 and Fig. 4a). The super-cooling of the transition from A to B, $T_{AB}/T_c$, and its pressure

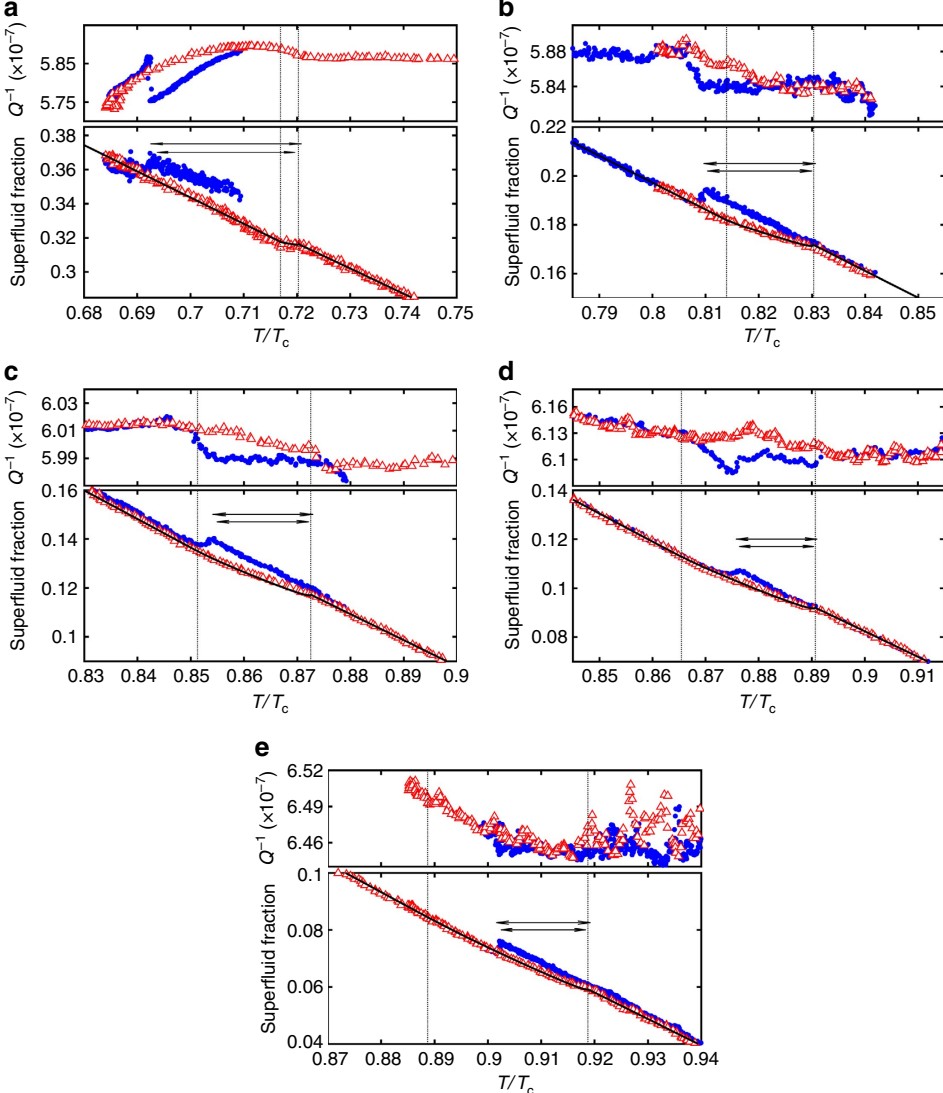

**Figure 3 | Superfluid fraction and dissipation at the A–B transition.** Measured superfluid fraction and dissipation ($Q^{-1}$) near the A–B transition at (**a**) 0.1 bar, (**b**) 1.4 bar, (**c**) 2.5 bar, (**d**) 3.6 bar and (**e**) 5.6 bar. The dotted vertical lines mark the onset $T_{BA}^{lower}/T_c$ (left) and completion $T_{BA}^{upper}/T_c$ (right) of the B–A transition on warming (open red triangles). This gradual transition is associated with the pressure-induced cavity height distribution; solid black lines show fits to the data assuming the transition at constant reduced thickness $D/\xi_\Delta(T_{AB})$ see Supplementary Note 1 for details. On cooling (filled blue circles) the B phase nucleates abruptly from the supercooled A phase. At 2.5–5.6 bar, the supercooling is smaller than the width of the warming transition, and the jump corresponds to a transition from pure A phase into spatially separated A/B phase coexistence. The dissipation increases across the A–B transition, particularly at low pressures. Horizontal arrows designate the range of the mean supercooling, $\delta T/T_c$, measured at that pressure. Values for various relevant quantities are listed in Table 1.

dependence are also shown in Table 1 and Fig. 4a. Finally, the ratio of the superfluid densities at this transition are also shown in Fig. 4b and listed in Table 1. We describe the systematics of these data in turn, followed by a discussion.

**Warming transition.** The warming transition $T_{BA}$, determines the thermodynamic transition temperature, as confirmed by the lack of history dependence of this feature (Fig. 5a). However, this transition exhibits a finite width, because of the smooth variation in height across the annular cell's cross-section, arising from pressure-induced bowing, as in ref. 36. The cross-section of the annular region under pressure is shown in Fig. 1c,d and Supplementary Fig. 3, (also Methods).

At each pressure we identify the start $T_{BA}^{lower}$ and end temperature $T_{BA}^{upper}$ of the transition (see details in Supplementary Note 1) determining the transition's width, $\Delta T/T_c = (T_{BA}^{upper} - T_{BA}^{lower})/T_c$.

The increase in transition width with pressure, (Table 1 and Fig. 4a), is consistent with the calculated bowing. $T_{BA}/T_c$ increases with increasing pressure, driven by the decrease of the zero temperature coherence length; a comparison with theoretical prediction is made later.

**Supercooling.** The transition at $T_{AB}$ while cooling (blue circles Fig. 3a–e) is indicated by a small but abrupt jump in superfluid fraction (Fig. 4b and Table 1). The apparent width (in temperature) of this transition is limited by the cooling rate $\sim 10\,\mu K.hr^{-1}$ and the oscillator decay time ($\sim 1000\,s$). The supercooling of the A phase $\delta T/T_c = (T_{BA}^{upper} - T_{AB})/T_c$ is shown in Table 1. It is measured from the upper temperature (completion) of the B→A transition, since this corresponds to maximum cavity height, where the B phase is nucleated on cooling. The supercooling is

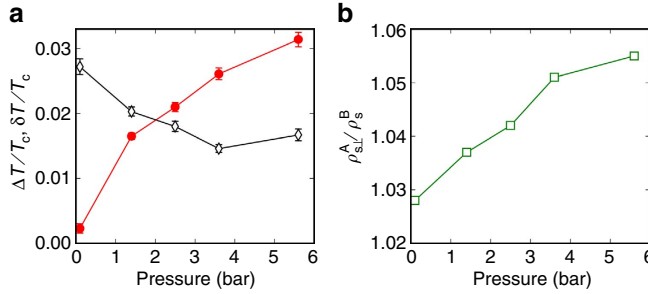

**Figure 4 | Pressure dependence of the measured properties of the A–B transition.** (**a**) The width of the B–A transition, $\Delta T/T_c$, (due to the pressure-induced bowing of the cavity) observed on warming (filled red circles) is compared to the extent of supercooling, $\delta T/T_c$, (open black diamonds), which shows a non-monotonic behaviour. Lines are guides to the eye. Error bars represent s.e.m. (see Table 1 for details). (**b**) The ratio $\rho_{s\perp}^A/\rho_s^B$ of superfluid fractions in the A and B phases near the transition; the departure from unity indicates the presence of the strong coupling effects down to low pressure.

extremely small in comparison with that observed in bulk, and exhibits a non-monotonic pressure dependence, Fig. 4a.

**Absence of pinning.** The finite width of the BA transition observed on warming, attributed to cell bowing, implies that during this transition the AB interfaces in the cell are positioned such that $D/\xi_\Delta(T_{AB})$ is at the critical value (see Supplementary Note 1). The surfaces of the cavity were polished (see Methods) in an attempt to eliminate pinning. To confirm the absence of pinning the experiment was warmed partially into the B→A transition region and then re-cooled in turnarounds, Fig. 5. The data clearly shows no hysteresis between warming and cooling and therefore no pinning. Warming to just above $T_{BA}^{upper}$ and subsequent cooling well reproduces the 'supercooled' trajectory. This behaviour contrasts to the data obtained in the 680 nm cavity that showed hysteresis associated with pinning of the A-B interface at scratches on the glass surface of that cavity[36,37]. The supercooling and warming transitions are depicted schematically in Fig. 5b,c.

**Superfluid density and dissipation.** The ratio of the superfluid density in the A phase and the corresponding value in the B phase is precisely determined at B phase nucleation, $T_{AB}/T_c$, Table 1. We also note that the dissipation is greater in the B phase (Fig. 3a–e) than in the A phase. In the bulk, the reverse is true[58]. When the A and B phases co-exist, the dissipation in this coexistence region is consistent with a contribution from each of the A and B phases. Thus, there is no identifiable additional dissipation associated with the presence of the A-B interface where a Kelvin–Helmholtz instability[59] may contribute. We

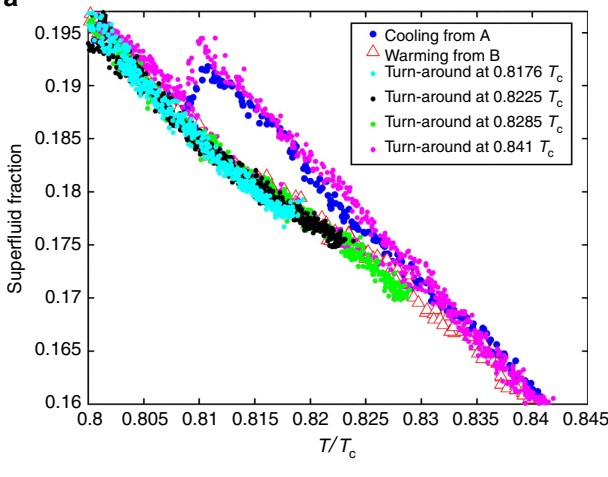

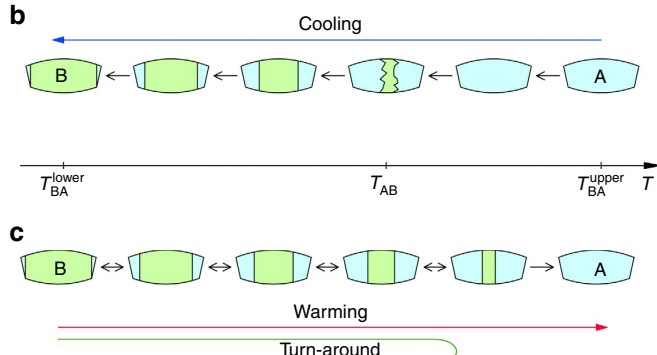

**Figure 5 | Traversal of the A–B transition.** (**a**) Superfluid fraction on different trajectories through the A–B transition region at 1.4 bar. (**b,c**) Schematic depiction of the distribution of the A and B phases in the cavity in the presence of bowing. (**b**) The A phase supercools down to $T_{AB}$, where the sample converts into the B phase at once. For 1.4 and 0.1 bar $T_{AB}$ occurs below $T_{BA}^{lower}$. At 2.5, 3.6 and 5.6 bar, $T_{AB}$ occurs above $T_{BA}^{lower}$ and the transformation from A to B phase continues on cooling below $T_{AB}$ till the B phase fills the annular cavity at $T_{BA}^{lower}$ with the exception of the tapered edges (see Supplementary Figure 2), which remain in the A phase due to strong confinement. (**c**) These edges serve as seeds of the A phase at the gradual B–A transition on warming via the A/B coexistence state. If the sample is cooled again in a turnaround from the coexistence state, the A to B transition also occurs gradually, following the route of the warming transition, indicating that on warming the equilibrium transition is observed.

observe that the dissipation excess at $T_{AB}$ is pressure-dependent and decreases as the pressure increases. This may be indicative of a contribution to the dissipation from surface Andreev bound states in the B-phase, confined within distance $\sim \xi_0$ from each wall, where $\xi_0$ decreases with pressure[60].

**Table 1 | Measured properties of the A–B transition in a 1.08 μm slab.**

| P (bar) | n | $\rho_{s\perp}^A/\rho_s^B(T_{AB})$ | $T_{AB}/T_c$ | $T_{BA}^{lower}/T_c$ | $T_{BA}^{upper}/T_c$ | $\delta T/T_c$ (supercooling) | $\Delta T/T_c$ (width) | $D/\xi_\Delta(T_{BA}^{lower})$ |
|---|---|---|---|---|---|---|---|---|
| 0.1 | 1 | 1.028 | 0.693 ± 0.0006 | 0.718 ± 0.0022 | 0.7202 ± 0.001 | 0.0272 ± 0.0012 | 0.0023 ± 0.0012 | 10.43 ± 0.03 |
| 1.4 | 4 | 1.037 | 0.8125 ± 0.0016 | 0.8163 ± 0.0043 | 0.8328 ± 0.0018 | 0.0203 ± 0.0007 | 0.0165 ± 0.0021 | 10.66 ± 0.10 |
| 2.5 | 5 | 1.042 | 0.8597 ± 0.0025 | 0.8567 ± 0.0039 | 0.8777 ± 0.0026 | 0.0180 ± 0.0008 | 0.021 ± 0.0033 | 11.05 ± 0.13 |
| 3.6 | 4 | 1.051 | 0.8803 ± 0.0018 | 0.8677 ± 0.0032 | 0.8938 ± 0.0019 | 0.0146 ± 0.0006 | 0.0261 ± 0.0027 | 12.02 ± 0.13 |
| 5.6 | 2 | 1.055 | 0.9014 ± 0.0014 | 0.8865 ± 0.002 | 0.9179 ± 0.0006 | 0.0167 ± 0.0009 | 0.0314 ± 0.0031 | 13.38 ± 0.11 |

The superfluid fraction ratio $\rho_{s\perp}^A/\rho_s^B(T_{AB})$ at the A–B transition, the mean temperature $T_{AB}/T_c$ of the A–B transition on cooling, together with the temperature $T_{BA}^{lower}/T_c$ of the start (lower end) of the B to A transition on warming and $T_{BA}^{upper}/T_c$ (upper end) where all the B phase is converted to the A phase are presented. The extent of supercooling $\delta T/T_c = (T_{BA}^{upper}/T_c - T_{AB})/T_c$ and the width of the B–A transition $\Delta T/T_c = (T_{BA}^{upper}/T_c - T_{BA}^{lower}/T_c)$ are computed from values in each of the $n$ complete warming and cooling cycles at various pressures. These are then averaged and the mean and s.e.m. of $\delta T/T_c$, and $\Delta T/T_c$ computed. The best fit reduced thickness of the warming transition $D/\xi_\Delta(T)$ at $T_{BA}^{lower}$ is also listed. In this temperature range, we measure the thermometry noise to be $\pm 0.0004$ mK. The errors are listed are the s.e.m. values calculated from all available crossings through the A–B transition.

## Discussion

In bulk, the conventional homogeneous nucleation theory predicts the lifetime of the supercooled A phase to exceed the age of the universe[61]. The nature of the mechanism for the nucleation of the B phase remains a matter of debate with several competing scenarios (Baked Alaska[41,43], Kibble-Zurek[44], Q balls[46] and resonant tunnelling[47]). Several of these scenarios rely on an extrinsic mechanism, in which local heating of the superfluid is caused by the energy deposited by an incoming particle (for example, neutron or cosmic ray). A threshold for the A–B transition has been reported that is consistent with the Baked Alaska model, but might be associated with an intrinsic process[62,63].

Under confinement in our thin slab geometry, we observe only very small supercooling, less than $0.03T_c$. By contrast, in bulk (at 33.6 bar) with clean surfaces, the observed lifetime of the supercooled A state is exceedingly long at temperatures near the equilibrium AB transition (that is, the B phase does not nucleate). Only by cooling to very low temperatures ($\cong 0.25T_c$, far below the equilibrium AB transition), is the transition to the B phase observed while holding the temperature fixed for several hours. Our experimental practice involves cooling at a steady rate through the supercooled state for several hours; such rates would result in supercooling in the bulk to $\cong 0.4T_c$ (ref. 38).

The critical radius of a bulk B phase nucleation bubble at $T = 0$, $R_0$, is inferred from experiment to be of the order of 0.5 µm at high pressures, and in zero magnetic field scales approximately as $R(T) = R_0(1 - T/T_c)^{1/2}/(1 - T/T_{AB})$, diverging at the equilibrium $T_{AB}$, where the free energy difference between the two phases vanishes[38]. We infer from measurements of the surface energy at $P = 0$ (ref. 64), and other thermodynamic data, that $R_0$ is comparable at low pressures. Since, under confinement, supercooling is small, $R(T)$ is much greater than $R_0$ and therefore $D$; the relevant nucleation volume is a disc of height $D$ and radius $R$, ruling out homogeneous nucleation due to its macroscopic size.

The sample geometry of our cavity is particularly well suited to studies of B phase nucleation. When the cell is bowed under pressure the B phase is expected to nucleate near the thickest region of the slab, where the B phase is the lowest free energy state (see Supplementary Note 2). Experimentally it is clear that, at the three higher pressures, $T_{AB} > T_{BA}^{lower}$. This demonstrates that nucleation occurs in the thicker part of the slab, surrounded by the more confined edges of the annulus where the sample remains in the A phase to lower temperature. The edges act as a bottle isolating the interior region, where B phase nucleation occurs, from any B phase otherwise present (for example in the fill line). This is functionally equivalent to the small 0.6 T NdFe permanent magnets that were used in experiments at Stanford[38]. There the locally strong magnetic field stabilized the A phase creating a valve to isolate the helium under study in quartz tubes from bulk $^3$He-B nucleated in a sinter or other poorly characterized material. Our cell's surface roughness is well characterized and much smoother than the quoted value ($< 10$ nm) in the Stanford experiments[38]. This is relevant since surface roughness is also implicated in the nucleation process. The dependence of the A phase supercooling on the sample history has been observed in the bulk[65]. In our experiment no memory effects are present with exception of the behaviour upon partial traversals of the transition (Fig. 5). Nucleation is also thought to be initiated by mechanical shocks applied to the experiment[65]. Our torsion oscillator (TO) is particularly sensitive to such impulses and the absence of additional noise at $T_{AB}$ argues against the premature nucleation of the B phase by accident. We did not administer such impulses to test the sensitivity to mechanical shocks.

The smallness of the supercooling and the smallness of the sample volume, strongly suggest that the B-phase nucleation we observe under confinement is an intrinsic phenomenon. We note that the AB transition in aerogel, by contrast, shows large supercooling[39,40] which suggests that surfaces by themselves are not so important. A scenario that is consistent with the small supercooling and draws on the putative stripe phase is the Tye–Wohns mechanism[47], which invokes the presence of intermediate hypothetical states between the A phase and the B phase which aids nucleation via resonant tunnelling. Under confinement, a natural intermediate state exists: the stripe phase. However, strong coupling corrections are predicted to strongly influence the stability of the equilibrium phase[22]. Two possible scenarios are therefore that over some region of the phase diagram the stripe phase is stable, and the A phase makes a first-order transition into the stripe phase which continuously evolves into the B-phase or that the stripe phase is not stable but provides the required nearby set of quantum vacua (for example via resonant tunnelling or another competing mechanism) to nucleate the B phase (see Supplementary Note 3).

The calculated phase diagram[22] is shown in Fig. 6, determined from GL theory, which strictly applies in the $T \to T_c$ limit, and is based on the strong coupling $\beta$-parameters proposed by Choi et al.[66]. This calculation applies a correction to standard GL theory, which takes into account a linear scaling in temperature of the strong coupling correction to the weak coupling $\beta$-parameters, and successfully reproduces the experimental bulk phase diagram at high pressures. These strong coupling parameters do not extrapolate to the weak coupling limit at zero pressure, consistent with earlier work which concluded that strong coupling corrections continue to play a role at zero pressure[67]. With these parameters the stripe phase is restricted to a thin sliver of the pressure-temperature phase diagram between 1 and 3 bar (ref. 22; Fig. 6) in contrast to a significant wedge predicted on the basis of GL theory with $\beta$-parameters derived from theory (Supplementary Fig. 4)[22,68]. The region of stability of the stripe phase, if any, is thus an open question. In the present experiment, the observed minimum in the extent of the supercooling of the A phase (Fig. 4) is well aligned with the predicted region shown. It is suggestive that the minimum in supercooling may arise from the virtual presence of the stripe phase to mediate the nucleation of the B phase. To test the resonant tunnelling scenario one would have to quickly traverse the region where the stripe phase is marginally stable and examine the statistics of nucleation (see Supplementary Note 3). We were unable to carry out rapid cooling strategies or measurements at high pressures due to technical factors.

The growth of the A phase from the B phase (Fig. 5), is not subject to any nucleation barrier. There are small features at the side walls of the cavity, where the A phase (favoured because of confinement) is still present even after cooling deep below the observed completion of the A $\to$ B transition (Supplementary Fig. 2). We follow a procedure (see Supplementary Note 1) to determine a best fit value at the thermodynamic B $\to$ A transition for $D/\xi_\Delta(T)$ at each pressure and plot the best fit for the measured $\rho_s/\rho$ versus $T/T_c$ in Fig. 3 as the solid line. The locations in $T/T_c$ of the start of the B $\to$ A transition ($T_{BA}^{lower}/T_c$) and the end of the B $\to$ A transition ($T_{BA}^{upper}/T_c$) along with $D/\xi_\Delta(T)$ at each pressure are also listed in Table 1.

The measured phase diagram, in comparison with the predicted phase diagram[22] for this cavity height, is shown in Fig. 6. To test the universality of $D/\xi_\Delta(T_{BA})$ we also compare the present measurements on a 1,080 nm cavity, using the superfluid density, and the previous NMR experiment on a 680 nm cavity (Fig. 6b–d)[36,37]. There is good agreement of $D/\xi_\Delta(T_{BA})$ between the two experiments (Fig. 6b). In weak coupling theory $D/\xi_\Delta(T_{BA})$ is pressure independent and depends only weakly on specularity of surface scattering[55–57], (Fig. 6b). Thus we confirm that strong coupling corrections play a role at all pressures. The most recent theory, Fig. 6[22,69], still shows significant discrepancies with the data, emphasizing the uncertainty in our current knowledge of the strong coupling parameters at low

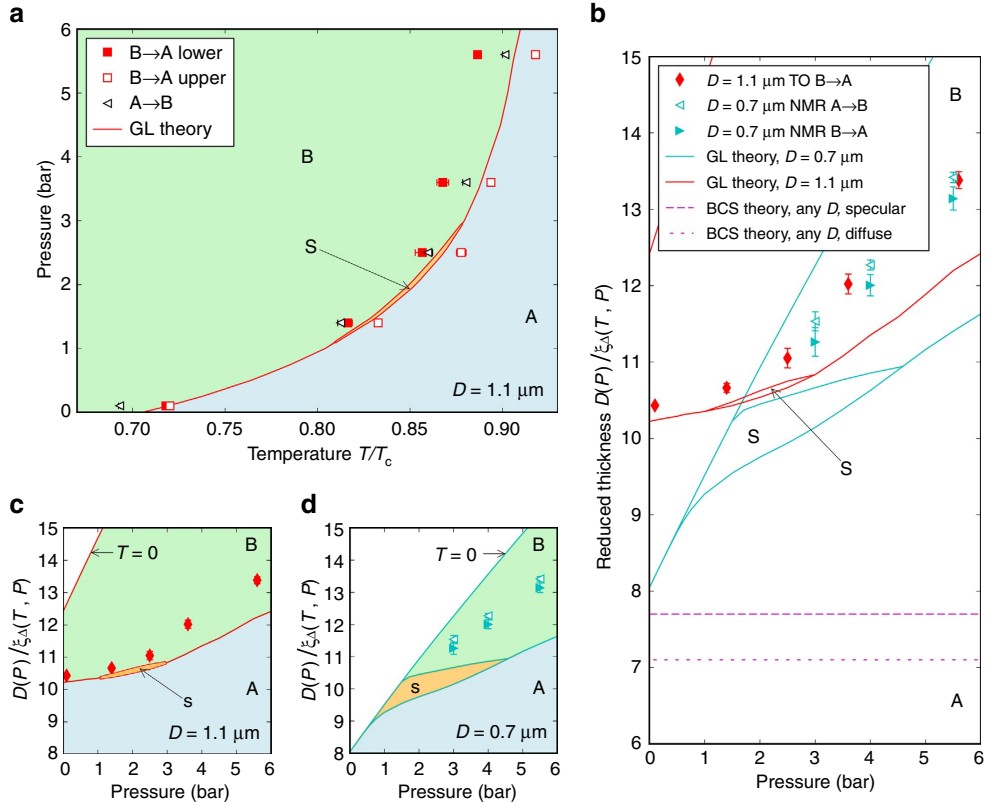

**Figure 6 | Phase diagram of superfluid $^3$He confined in slab geometry. (a)** The A–B transition in the $D = 1.1\,\mu m$ slab (this experiment) in the temperature–pressure plane. The start, $T_{BA}^{lower}$ (filled squares), and completion, $T_{BA}^{upper}$ (open squares), of the B–A transition on warming correspond to the equilibrium B–A transition in the thinnest (fixed $D = 1.08\,\mu m$ near the edge) and thickest (pressure-dependent $D$) parts of the cavity, respectively. On cooling the A phase supercools down to $T_{AB}$ (black triangles). Solid red lines and blue/orange/green areas show the A/stripe-S/B phase diagram predicted by the GL theory[22] with experimentally determined strong coupling parameters for $D = 1.08\,\mu m$, therefore directly comparable to $T_{BA}^{lower}$. The A phase is observed in the predicted region of stability of the stripe phase (orange), demonstrating the inaccuracy of the strong coupling parameters used for the calculation[22]. **(b)** The reduced thickness $D/\xi_\Delta$ representation of the phase diagram allows comparison of this TO experiment (in this representation $T_{BA}^{lower}$ and $T_{BA}^{upper}$ coincide) with the NMR experiment on a $D = 0.7\,\mu m$ slab[36,37] (due to hysteresis the equilibrium transition was not observed, here the data on warming and cooling are shown). The two data sets' collapse demonstrates the universality of $D/\xi_\Delta$ at the A–B transition. These measurements strongly deviate from the prediction of the weak coupling (BCS) theory, shown here for diffusely and specularly scattering cavity walls[57]. Since the nature of boundary scattering only weakly affects coordinates of the BCS A–B interface, we compare the experiments with diffuse walls to the predictions of the GL theory only available for specular walls[21,22] (same in **a**). The phase boundaries derived within the GL theory for $D = 1.08\,\mu m$ and $0.7\,\mu m$ (red and cyan lines in **b**), also shown separately in (**c**) $D = 1.08\,\mu m$ and, (**d**) $D = 0.7\,\mu m$ depart from the $D/\xi_\Delta$ collapse due to the temperature dependence of the strong coupling included in the theory[21,22]. The disagreement between the experiments and GL theory emphasizes the current limited understanding of the strong coupling parameters at low pressure that leaves the stability of the stripe phase uncertain. Error bars represent s.e.m.

pressure. The theoretical lines for $D = 0.7\,\mu m$ and $1.1\,\mu m$ are offset due to the proposed temperature dependence of the strong coupling corrections, so the reduced thickness $D/\xi_\Delta(T_{BA})$ at the phase boundary is no longer universal. The shift between the data from the two cavities is small. Stronger discrepancy between experiment and theory at $0.7\,\mu m$ than at $1.1\,\mu m$ suggests that the theory underestimates the residual strong coupling at low temperature. This plot also highlights the sensitivity of the putative equilibrium stripe phase to the precise details of pressure-dependent strong coupling parameters and illustrates the pressure window over which a nearby stripe phase may play a role in B phase nucleation.

Due to bowing, the A–B interface will be oriented azimuthally, and hence parallel to the flow. Thus the putative stripe phase would be located at this interface, parallel to the flow with expected minimal effect on the superfluid fraction in our experiment. The present setup cannot rule out a thermodynamically stable stripe phase.

The observed superfluid density of the A phase at $T_{AB}$ is greater than that of the B phase (for $\rho_{s\perp}^A/\rho_s^B(T_{AB})$ see Table 1,

Figs 3 and 4b). The component of the superfluid density tensor assayed is $\rho_{s\perp}^A$ (ref. 70), since the $\hat{\ell}$ vector is oriented perpendicular to the surfaces[71]. $\rho_{s\perp}^A/\rho_s^B(T_{AB})$ decreases as the pressure is lowered (Table 1 and Fig. 4b). Although the B phase gap is subject to a planar distortion due to confinement, the superfluid density is sensitive to the in-plane component of the gap, which should be close to the bulk isotropic value[54]. Detailed calculations of the superfluid density under confinement are however required. Since the ratio of superfluid fractions is unity in the weak coupling limit and our data always shows that $\rho_{s\perp}^A/\rho_s^B > 1$, this measurement provides further support for the presence of strong coupling even at the lowest pressures[72].

In conclusion, we have made a torsional pendulum study of superfluid $^3$He confined in a single micron-sized cavity. The ease with which the B phase nucleates provides clear evidence for an intrinsic nucleation mechanism at the first-order A→B transition under such confinement. We established, via the scaling of the equilibrium AB transition temperature with cavity height, that strong coupling effects at low pressures are stronger than currently believed. This has implications for the stability of the

stripe phase, which was not directly observed in this experiment. A possible mechanism for the efficient nucleation of the B phase invokes resonant tunnelling where even if not stable, the stripe phase with its spatially modulated order parameter can provide the intermediate metastable states to mediate the transition. Since the energetics of the stripe phase appear be particularly sensitive to the choice of cavity height and pressure, the detailed systematics of B phase nucleation, in response to temperature and pressure changes for different cavity height, should provide a further test of the potential nucleation scenarios. Nanofluidic cells also offer the potential to control and manipulate the A–B interface via stepped size modulation.

It is believed to be likely that first-order phase transitions occurred in the early universe, may explain matter–antimatter asymmetry, and are important in inflation scenarios. According to standard nucleation theory, the supercooled A phase lifetime in bulk superfluid should greatly exceed the lifetime of the universe, and this has prompted proposals for a number of extrinsic nucleation mechanisms. The fact that supercooling is virtually eliminated under confinement is striking. The notion that confinement significantly modifies the energy landscape, creating false vacua that promote B phase nucleation, via resonant tunnelling or another competing mechanism, merits further study in the laboratory, which may impact on our understanding of these central questions in cosmology.

## Methods

**Cell construction.** The 14 mm diameter silicon disk that comprises the micro-machined chamber to contain the $^3$He was fabricated at Cornell's Nanofabrication facility[49]. The process flow is shown in the Supplementary Fig. 1. After patterning of the silicon a matching octagonal piece of highly polished Sodium-doped glass (Hoya SD-2) was bonded to the silicon. The so-constructed cavity comprises the head of the torsion pendulum (Fig. 1a,b), and was mounted to the coin silver torsion rod using a special alignment jig and Tra-Bond 2151 epoxy. The free volume in the torsion rod was minimized by using a quartz tube of 320 μm outer diameter and 100 μm inner diameter that also served to exclude the epoxy (because of its high viscosity) from flowing into the cavity.

**Characterization of cavity geometry.** The fabricated cavity height (1,080 nm) is maintained only at the bonded points of attachment. The step height on the cavity was measured prior to bonding with a Tencor P10 profilometer and found to be 1,080 nm. Roughness of the silicon cavity surface was measured using an atomic force microscope to be $0.102 \pm 0.035$ nm (arithmetic average, $R_a$, and s.d.). The same measurement for the glass surface after polishing was measured to be $0.342 \pm 0.049$ nm (arithmetic average, $R_a$, and s.d.).

Finite element analysis (Fig. 1c,d) and Supplementary Fig. 3 reveals the extent to which walls of the rectangular cross-section nanofluidic cavity bow under pressure. We modelled the bowing using finite element methods (using COMSOL multiphysics software) and materials properties from standard tables that produce an expected bow (at maximum) of 30 nm bar$^{-1}$. Measurements on a different cavity made from the same glass and silicon and having geometry suitable for NMR investigations yield a best fit of 32 nm bar$^{-1}$ bowing. The fits shown in Fig. 3 use the 31 nm bar$^{-1}$ figure and the error bars for $D/\xi_\Delta(T_{BA}^{lower})$, $T_{BA}^{lower}/T_c$ and the resulting $\Delta T/T_c$ in Table 1 and figures, were obtained taking into account the difference between the calculated and experimental figures. Error bars for $T_{AB}/T_c$, $T_{BA}^{lower}/T_c$, $T_{BA}^{upper}/T_c$ and the resulting $\delta T/T_c$ are determined from the noise in the experimental data, and represent the s.e.m. from all $n$ available runs (Table 1), with the exception of 0.1 bar, where a single temperature cycle was performed and the fit errors are used instead.

The edge of the annular cell cavity retains features due to the fabrication process where small regions of a few μm width of A phase likely persist well below $T_{AB}$. These features are illustrated in Supplementary Fig. 2.

**Torsion pendulum.** Coin silver (90% silver and 10% copper) was chosen as the alloy for the torsion pendulum because this material provides a high $Q$ at low temperatures and has a relatively small temperature-dependent frequency background[73]. As with our usual practice we drove and detected the pendulum motion electrostatically, keeping it close to resonance using a digital phase locked loop. The resonant frequency of the mode we excited the pendulum at was $\sim 1,330$ Hz and the quality factor was $\sim 1.5 \times 10^6$ at mK temperatures. Below the superfluid transition, the superfluid fraction of the fluid in the cavity decouples from the pendulum, and a decrease in its period is observed. The ratio of the moment of inertia of the fluid in the cavity to the moment of inertia of pendulum head was $\sim 6$ parts in $10^6$. The frequency noise is $\sim 2$–4 parts in $10^9$ giving us ample resolution to resolve the superfluid fraction $\rho_s/\rho$. Achieving this degree of frequency stability required a compromise between driving

the pendulum at a large enough amplitude (to increase signal above ambient vibrational noise) and nonlinearity of the pendulum's torsional mode.

The superfluid fraction was determined from the frequency of the torsion pendulum, $f(T)$, through the following expression:

$$\frac{\rho_s}{\rho} = \frac{\Delta f(T)}{\Delta f_{fluid}} \qquad (1)$$

where $\Delta f(T)$ is the difference between the measured frequency $f$ and the frequency of the pendulum if all the fluid is fully locked $f_{full}$. At $T_c$ the viscous penetration depth of $^3$He is $\sim 0.5$ mm, many orders of magnitude larger than the distance between the plates. We can assume that for temperature slightly above $T_c$ to $T_c$ all the fluid is fully coupled to the pendulum and $f_{full} = f$. For values of $f_{full}$ below the superfluid transition, we extrapolate from the normal state values. $\Delta f_{fluid}$ is equal to the difference in the pendulum frequency between empty and filled. $\Delta f_{fluid}$ should scale linearly with the density of the fluid. When the sample is pressurized, the axis of the pendulum distorts slightly. Below 1 K, $^4$He is nearly 100% superfluid, so any difference between the filled cell and empty cell frequency below 1 K will be only due to torsion rod distortion. By comparing empty cell data and data for $^4$He at $T < 1$ K we determine that this effect was responsible for a frequency shift of 1.78 mHz for 3 bar of pressure. After the effect of torsion rod distortion is accounted for, we observe that the difference for the torsion pendulum frequency between fully coupled fluid just above $T_c$ and the empty cell frequency is 7 mHz. Using the appropriate values for the density of $^4$He at these temperatures and pressure, we determine that the expression for the frequency shift due to the fully coupled normal fluid is:

$$\Delta f_{fluid} = 46.05 \left[ mHz \cdot cm^3 \cdot g^{-1} \right] \times \rho \left[ g \cdot cm^{-3} \right] \qquad (2)$$

where $\rho$ is the density of $^3$He at the superfluid transition.

**Data availability.** The data that supports this study is available through Cornell University e-commons data repository at http://hdl.handle.net/1813/46294

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

## Acknowledgements

We acknowledge input from H. Tye, M. Perelstein, E. Mueller, J.A. Sauls, J.J. Wiman, Hao Wu and A.Casey. We also acknowledge the assistance of R. DeAlba with imaging of the silicon surface depicted in Supplementary Fig. 2. This work was supported at Cornell by the NSF under DMR-1202991 and in London by the EPSRC under EP/J022004/1, and the European Microkelvin Platform.

## Author contributions

Experimental work and analysis was principally carried out by N.Z. assisted by T.S.A. with further support from E.N.S. and J.M.P. R.G.B. and N.Z. established the nanofabrication protocols, and X.R. and R.G.B. carried out the finite element modelling. L.L. analysed much of the data for comparison of the 680 and 1,080 nm data sets, and N.Z. and L.L. shared in much of the data analysis. J.M.P. supervised the work and J.M.P. and J.S. had leading roles in formulating the research and writing this paper.

## Additional information

**Competing interests:** The authors declare no competing financial interests.

