## [Peer Review File · Nature Communications]

Reviewers' comments:

Reviewer #1 (Remarks to the Author):

This is an excellent and extremely well-written report on novel experiments, some of the first fruits from an international collaboration that combines the expertise and knowledge of two of the world's leading laboratories studying condensed matter systems at ultra low temperatures. The authors report on measurements using a beautiful new tool that they have developed for studying the mechanical properties of superfluid condensates under extreme confinement.

When taking matter into completely new regimes, one might expect (hope?) to make new discoveries. Here, they have done just that, and report on the implications of their first measurements. With very careful and considered arguments they examine intriguing new evidence for the nucleation properties of the AB transition within the superfluid phases, perhaps shedding light on a long-standing mystery. This becomes possible here because under the right conditions confinement should promote a new "stripe" phase. The authors are searching for evidence for this phase, and have realised that it may act as a possible nucleation intermediary. This insight is compelling, and of great interest to researchers in superfluids and also in quantum fluid condensate systems in general, from exotic superconductors to cold atomic gases. Further, the parallels between the order parameter of superfluid helium-3 and the quantum vacuum state of the early universe enable the authors' measurements potentially to offer support to the cosmological resonant tunnelling model of Tye et al. This is exciting, and the authors go so far as to suggest future experiments that might actually test it.

It is also important to note the second main message of this work, that the results provide new evidence for strong coupling at lower pressures than generally accepted. This will undoubtedly influence thinking in the field for both future theoretical prediction and the analysis of experimental data.

I have only a few queries that the authors should address.

1. It is known (at least anecdotally) that mechanical shock can trigger nucleation of the B phase from the A phase. I think the authors' methods preclude this, but it would be nice to see a comment in the paper.
2. I cannot tell how the authors have derived the error bars in Fig. 4(a). Apologies if this should be obvious. It is important because the supercooling data is clearly non-monotonic in pressure within the error bars as drawn and forms part of the evidence for a new nucleation process. Do they represent the spread in the nucleation temperature for a single cooldown at each pressure, or an average over several cooldowns? For instance I see that there is more than one cooldown through nucleation at 1.4 bar, shown in Fig. 5 (which has a very clear cartoon of the turnaround process, by the way). A comment from the authors on the reproducibility/stochasticity of the supercooling would lend more experimental weight to their observation that the behaviour is non-monotonic in pressure.
3. The authors state that the nucleation of the B phase must take place where the cavity height is maximum. I think this is an assumption (though probably true); or is there some interpretation of the experimental data that proves it?
4. The resonant tunnelling model is not that widely known. The authors might consider adding a paragraph to supplemental material that explains the mechanism in a little more detail and what its predictions would be for the nucleation behaviour observed here.

Finally a minor point. The authors should be commended on the number of citations they have pulled together. But I noticed that references 31-33 are the same as 63-65. I should also mention

that there is experimental work from Helsinki and Lancaster on the AB transition and interface that the authors may (or may not) find interesting and consider to be relevant to their setup and interpretations. Eg.: Shear flow and Kelvin-Helmholtz instability in superfluids, Phys. Rev. Lett. 89, 155301 (2002) (relevant to coexistence of A and B in the torsional oscillator?); Primary and secondary nucleation of the transition between the A and B phases of superfluid helium-3, Phys. Rev. Lett. 85, 4321 (2000) (more evidence that Baked Alaska can't explain all nucleation?); Relic topological defects from brane annihilation simulated in superfluid helium-3, Nat. Phys. 4, 46 (2008) (the AB interface itself as a cosmological analogue?).

In summary, this is original work of the highest quality at extreme levels of experimental difficulty, with conclusions that are well thought out and findings which I am sure will impact on a wider community than the core research area(s) of the authors.

Reviewer #2 (Remarks to the Author):

The manuscript describes torsional oscillator experiments on superfluid ^3He confined in a nano-cavity. The thickness is chosen as 1.08 micrometers to favor the conditions where they expect to observe effects related to the "stripe phase" predicted by Sauls, Vorontsov and Wiman.

The main results are the very small degree of supercooling observed for the A-B transition, which is attributed to the presence of the stripe phase, and evidence for strong coupling corrections being larger than expected in superfluid ^3He , is also obtained from the data.

The weak part of the analysis is, to my point of view, that one cannot clearly exclude other effects which could explain the absence of supercooling. Having said that, I have to admit that the authors have done everything they could do at this stage, to show that the stripe phase is the most reasonable option compatible with the experimental and theoretical results. Further measurements with different thicknesses and other materials will be needed, but would represent such an additional work, that it should be left to later works.

On the other hand, the conclusions are novel and interesting, they will generate substantial activity in the field of confined fermionic superfluids.

To conclude, I would say that the paper presents sound research, speculative to some degree, but certainly motivating and with consequences for a broad range of topics, as demonstrated by the variety of the subjects in the references.

I recommend publication of the manuscript in Nature Communications.

Reviewer #3 (Remarks to the Author):

It is certainly true that superfluid phases of ^3He are probably the most important of condensed matter systems due to their impact on superconductivity, cosmology, turbulence, particle physics and especially topological phases of matter - a topic in modern physics, which received in 2016 the Nobel Prize. That is why the new developments in the investigation of the superfluid ^3He are interesting to a wide audience of Nature Journals. The authors considered superfluid ^3He in a precisely engineered nanofabricated geometry, which allowed them to study in the limit of strong confinement the phase transition between two topological phases: the chiral A-phase and the B-phase, which is the superfluid analog of the time reversal invariant topological insulator. This is real progress in experimental studies of ^3He .

Nevertheless, the results are not very exciting and in fact they have no relations to the topological properties of these topological liquids. The measured phase diagram showed no fundamentally new features, such as the existence of the predicted stripe phase. The new result is that the observed supercooling of the A-phase appeared to be very small compared to that in the bulk ^3He , that may be relevant to the old and not very well understood problem of the B-phase nucleation. The authors propose an explanation in terms of the metastable stripe phase, which mediates the resonance tunneling between the phases A and B. However, the tunneling processes in superfluid

phases of ^3He are suppressed by many orders of magnitude. On the other hand, the observed effect is not very surprising: in confined geometry the pockets, where the B-phase nucleation is easy, cannot be excluded.

In conclusion, the experiments provide new knowledge in the study of superfluid ^3He , but lack enough novelty, which would justify the publication in Nature Communications.

Dear Referees

We thank you all for your comments and time in reading and assessing our manuscript, and are pleased that two reviewers recommend publication in Nature Communications. While reviewer 3 is convinced that “the superfluid phases of ^3He are probably the most important of condensed matter systems”, the reviewer has some criticisms which we address below. We also answer the concerns on the novelty and implications of our work, raised by this reviewer.

Below we separately supply the detailed response to each of the comments of each reviewer.

We would like to emphasize that our observations of B-phase nucleation, in the new regime of nanoscale confinement, are consistent with the resonant tunneling model; the RTM remains the best explanation for the observed supercooling.

We agree with Reviewers 1 and 2, who both support publication of this work, that a further exploration of the detailed systematics of supercooling in confined geometry is both exciting and needs to be pursued and we are actively planning a new experiment to explore this area. We are confident that when published, our experimental result will prove to be a catalyst for further experimental and theoretical work. Our hope is that referee 3, after seeing the more detailed discussion of the RTM and the responses will have a better explanation of the importance of understanding supercooling and its likely association with the putative stripe phase and will come to a revised opinion.

We hope, given our responses, that you will agree that the paper should move forward in the publication process.

Reviewer #1 (Remarks to the Author):

This is an excellent and extremely well-written report on novel experiments, some of the first fruits from an international collaboration that combines the expertise and knowledge of two of the world's leading laboratories studying condensed matter systems at ultra low temperatures. The authors report on measurements using a beautiful new tool that they have developed for studying the mechanical properties of superfluid condensates under extreme confinement.

When taking matter into completely new regimes, one might expect (hope?) to make new discoveries. Here, they have done just that, and report on the implications of their first measurements. With very careful and considered arguments they examine intriguing new evidence for the nucleation properties of the AB transition within the superfluid phases, perhaps shedding light on a long-standing mystery. This becomes possible here because under the right conditions confinement should promote a new “stripe” phase. The authors are searching for

evidence for this phase, and have realised that it may act as a possible nucleation intermediary. This insight is compelling, and of great interest to researchers in superfluids and also in quantum fluid condensate systems in general, from exotic superconductors to cold atomic gases. Further, the parallels between the order parameter of superfluid helium-3 and the quantum vacuum state of the early universe enable the authors' measurements potentially to offer support to the cosmological resonant tunnelling model of Tye et al. This is exciting, and the authors go so far as to suggest future experiments that might actually test it.

It is also important to note the second main message of this work, that the results provide new evidence for strong coupling at lower pressures than generally accepted. This will undoubtedly influence thinking in the field for both future theoretical prediction and the analysis of experimental data.

We also believe it is important to emphasize that there is a disparity between the experimental observation of T_{BA} and the corresponding theoretical predictions. We have clarified this by modifying our statement on page 12 and in the caption to Fig 6.

I have only a few queries that the authors should address.

1. It is known (at least anecdotally) that mechanical shock can trigger nucleation of the B phase from the A phase. I think the authors' methods preclude this, but it would be nice to see a comment in the paper.

Action: We have included a reference to the potential role of mechanical shock in the discussion on page 9 of the revised manuscript.

2. I cannot tell how the authors have derived the error bars in Fig. 4(a). Apologies if this should be obvious. It is important because the supercooling data is clearly non-monotonic in pressure within the error bars as drawn and forms part of the evidence for a new nucleation process. Do they represent the spread in the nucleation temperature for a single cooldown at each pressure, or an average over several cooldowns? For instance I see that there is more than one cooldown through nucleation at 1.4 bar, shown in Fig. 5 (which has a very clear cartoon of the turnaround process, by the way). A comment from the authors on the reproducibility/stochasticity of the supercooling would lend more experimental weight to their observation that the behaviour is non-monotonic in pressure.

Action In the original submission the error bars were based on the estimated uncertainties in individual measurements, selected as the most clear temperature sweep runs at each pressure. In response to the reviewer, all the data was re-examined. We now present the results of this new analysis. The errors quoted are the standard error of the mean from the available temperature sweeps.

Within our errors the non-monotonicity of supercooling with pressure is a robust result, although not quite as pronounced. We refer in text to evidence for a non-monotonic pressure dependence, which reinforces the observation of very small supercooling as evidence for RTM nucleation.

We have revised the table 1 and figures 3, 4 and 6. We include a discussion on errors as part of the supplemental note 1, section 10, of the revised manuscript. Indeed a more extensive study of the stochasticity is desirable. The detailed data for all sweeps used is included in the data repository.

3. The authors state that the nucleation of the B phase must take place where the cavity height is maximum. I think this is an assumption (though probably true); or is there some interpretation of the experimental data that proves it?

Action/Response: We have included a revised discussion on page 9 and a discussion in Supplemental note 2, referred to on page 9. This model relies on the dependence of the equilibrium BA transition temperature on effective confinement D/ξ_0 , discussed extensively in the manuscript. Weaker confinement favors the B phase. The nucleation of B phase in the most bowed region of the cavity is also supported by the NMR results on the AB transition reported in ref. 18.

4. The resonant tunneling model is not that widely known. The authors might consider adding a paragraph to supplemental material that explains the mechanism in a little more detail and what its predictions would be for the nucleation behaviour observed here.

Action: We have complied with this request and include a discussion in Supplemental note 3 which we refer to on pages 10, 11.

Finally a minor point. The authors should be commended on the number of citations they have pulled together. But I noticed that references 31-33 are the same as 63-65. I should also mention that there is experimental work from Helsinki and Lancaster on the AB transition and interface that the authors may (or may not) find interesting and consider to be relevant to their setup and interpretations. Eg.: Shear flow and Kelvin-Helmholtz instability in superfluids, Phys. Rev. Lett. 89, 155301 (2002) (relevant to coexistence of A and B in the torsional oscillator?); Primary and secondary nucleation of the transition between the A and B phases of superfluid helium-3, Phys. Rev. Lett. 85, 4321 (2000) (more evidence that Baked Alaska can't explain all nucleation?); Relic topological defects from brane annihilation simulated in superfluid helium-3, Nat. Phys. 4, 46 (2008) (the AB interface itself as a cosmological analogue?).

Action: We thank the referee for bringing these to our attention. We have added the Nature Physics paper to our list of papers reporting on the study of superfluid ^3He as a cosmological model. It appears as new Ref 11. We also refer to the Kelvin-Helmholtz instability (new Ref 58) and refer to it on page 7. The Primary and secondary nucleation (memory effect) is referred to in the main text page 9 and as Ref 64 as well as in the discussion on resonant tunneling in the supplemental note 3. We have deleted and reassigned the duplicate references. We have also added a reference to the exchange of comments (new ref 62) following the publication of Leggett's PRL (now Ref 61).

In summary, this is original work of the highest quality at extreme levels of experimental difficulty, with conclusions that are well thought out and findings which I am sure will impact on a wider community than the core research area(s) of the authors.

Reviewer #2 (Remarks to the Author):

The manuscript describes torsional oscillator experiments on superfluid ^3He confined in a nanocavity. The thickness is chosen as 1.08 micrometers to favor the conditions where they expect to observe effects related to the “stripe phase” predicted by Sauls, Vorontsov and Wiman. The main results are the very small degree of supercooling observed for the A-B transition, which is attributed to the presence of the stripe phase, and evidence for strong coupling corrections being larger than expected in superfluid ^3He , is also obtained from the data. The weak part of the analysis is, to my point of view, that one cannot clearly exclude other effects which could explain the absence of supercooling. Having said that, I have to admit that the authors have done everything they could do at this stage, to show that the stripe phase is the most reasonable option compatible with the experimental and theoretical results. Further measurements with different thicknesses and other materials will be needed, but would represent such an additional work, that it should be left to later works.

Action/Response: We agree that we cannot, at this time, conclusively prove that the stripe phase’s presence mediates the A to B transition. As stated by referee 2, below, our result on nucleation under confinement is novel and interesting. The additional discussion of the resonant tunneling model (in the supplemental note 3) and more discussion (in the same note) of the structure of the stripe phase as it pertains to the resonant tunneling model (RTM), added in response to comments of referee 1, should we hope be helpful here.

We agree that the task of designing an experiment (taking into account revised strong coupling parameters) to investigate the RTM and its relation to the striped phase is a significant and important task. We already briefly acknowledged this in our penultimate concluding paragraph. We plan to be undertake this in the near future. However, we believe (as stated by the referee’s concluding sentence) that the topic is broad, the results are sufficiently novel, that they should be published in a prestigious journal such as Nature Communications.

On the other hand, the conclusions are novel and interesting, they will generate substantial activity in the field of confined fermionic superfluids.

To conclude, I would say that the paper presents sound research, speculative to some degree, but certainly motivating and with consequences for a broad range of topics, as demonstrated by the variety of the subjects in the references.

I recommend publication of the manuscript in Nature Communications.

Reviewer #3 (Remarks to the Author):

It is certainly true that superfluid phases of ^3He are probably the most important of condensed matter systems due to their impact on superconductivity, cosmology, turbulence, particle physics and especially topological phases of matter - a topic in modern physics, which received in 2016 the Nobel Prize. That is why the new developments in the investigation of the superfluid ^3He are interesting to a wide audience of Nature Journals. The authors considered superfluid ^3He in a precisely engineered nanofabricated geometry, which allowed them to study in the limit of strong

confinement the phase transition between two topological phases: the chiral A-phase and the B-phase, which is the superfluid analog of the time reversal invariant topological insulator. This is real progress in experimental studies of ^3He .

Nevertheless, the results are not very exciting and in fact they have no relations to the topological properties of these topological liquids. The measured phase diagram showed no fundamentally new features, such as the existence of the predicted stripe phase. The new result is that the observed supercooling of the A-phase appeared to be very small compared to that in the bulk ^3He , that may be relevant to the old and not very well understood problem of the B-phase nucleation. The authors propose an explanation in terms of the metastable stripe phase, which mediates the resonance tunneling between the phases A and B. However, the tunneling processes in superfluid phases of ^3He are suppressed by many orders of magnitude. On the other hand, the observed effect is not very surprising: in confined geometry the pockets, where the B-phase nucleation is easy, cannot be excluded.

In conclusion, the experiments provide new knowledge in the study of superfluid ^3He , but lack enough novelty, which would justify the publication in Nature Communications.

We agree with the preamble (first paragraph) of Reviewer 3's comments. The excitement and interest of this work is recognized by reviewers 1 and 2.

Action/Response: The reviewer's central criticism "the observed effect is not very surprising: in confined geometry the pockets, where the B-phase nucleation is easy, cannot be excluded", while an obvious potential concern, is not well-founded for the three highest pressures where $T_{AB} > T_{BA}^{lower}$. As the manuscript explains, our bowed cavity geometry isolates the location of the B phase nucleation to the central region far away from the bulk B phase. This is discussed on p9 and illustrated in Fig. 5, and now further discussed in added supplemental note 2. The reviewer seems not to have appreciated that the A phase in these regions should be exponentially long lived (or conversely that nucleation should be exponentially suppressed), and the RTM due to the energetically nearby stripe phase provides the potential resolution of the relative ease with which B-phase nucleation occurs. We thank the reviewer for urging clarification of this important point.

Response: The referee states that the problem of B-phase nucleation is "old and not very well understood". Here we argue that this problem is important and of wide relevance, and our results offer new insights. In this manuscript we present evidence from the new regime of nanoscale confined superfluid, that an intrinsic nucleation mechanism (RTM), originally hypothesized in the context of bulk superfluid ^3He , should play a key role. The key qualitative new ingredient under confinement is the stripe phase, which should dramatically enhance tunneling rates. The relevance to models of the early universe is discussed in the concluding paragraph.

On relevance to topological superfluidity: The study of superfluid ^3He under nanoscale confinement offers the potential for significant new insights into the distinct topological properties of the A and B phases. These experiments represent a new and promising research direction. Mapping out the influence of confinement on the phase diagram, establishing the order

parameter, and the stability of potential new phases crucially underpins the study of these topological properties. It is particularly important to understand both interfaces between two phases with distinct topology (A and B), and the putative stripe phase, which itself will have interesting topological properties arising from its spatial modulation and domain wall structure. Here the broader relevance is that superfluid ^3He acts as a model for topological superconductivity.

Reviewers' comments:

Reviewer #1 (Remarks to the Author):

The authors have acted on all my recommendations and I certainly recommend publications in Nature Communications. I would reiterate that although the helium-3 nucleation problem has been known about for some time, there is still no satisfactory explanation. The results of this new study, and the RTM interpretation, may well prove to be a breakthrough, with significance for the understanding of phase transitions in general.

Reviewer #2 (Remarks to the Author):

The changes made to the manuscript clarify the points raised by the referees. In particular, the way the error bars were determined (explanation requested by referee 1) is now given in detail. Other corrections improve the paper, but they were not really essential. As referee 1 and myself clearly stated, the paper was extremely well written, and could have been published in its initial form.

The changes made to the manuscript clarify the points raised by the referees. In particular, the way the error bars were determined (explanation requested by referee 1) is now given in detail. Other corrections improve the paper, but they were not really essential. As referee 1 and myself clearly stated, the paper is extremely well written, and could have been published in its initial form.

The paper can be read at different levels. The introduction is excellent, the conclusions clearly described, the core of the paper and the supplementary material contain accurate descriptions of the experiments and the methods. Some parts can be found a bit heavy to digest for researchers not familiar with ^3He , but I find it appropriate to provide these details here. As said before, the subject touches a large variety of subjects, and the text has to provide sophisticated information adapted to different communities. I also appreciate particularly the fact that the results are very honestly described, and the same remark applies to the response to the referees.

I consider that the authors have provided adequate answers to the questions, comments and eventually the criticisms of all three referees. I strongly recommend the publication of the manuscript in Nature Communications.

Reviewer #3 (Remarks to the Author):

It is true that the experiment is really a high quality research: the phase diagram has been obtained under new condition of confinement. But from my point of view there are no new features. The observed lack of supercooling is not the fundamental result, while the suggested explanation of supercooling in terms of the resonant tunneling is certainly wrong. Resonant tunneling mechanism (RTM) applied to superfluid ^3He , even if it is possible, requires so strong fine-tuning, that can not be achieved in any real experiment with ^3He . However, the authors stress that the RTM is the best explanation for their observations. I can not buy such a statement. The authors should remove any relation to the RTM, if they want to stay in the area of real physics.

Reviewer #4 (Remarks to the Author):

The authors study a phase diagram of the superfluid ^3He confined in a slab geometry of 1.08 micrometer thickness. In the experiment, the theoretically predicted stripe phase is not detected

but the strong suppression of the supercooling of the A-B transition is observed. In order to explain the small supercooling, the authors propose the nucleation mechanism at the A-B transition by the resonant tunneling via the stripe state.

This experiment is high quality and sufficiently convince one that there is an intrinsic nucleation mechanism at the A-B transition. Then, this work sheds new light on a long-standing problem and is worth publishing in Nature Communications. I think that, however, the authors should not strongly propose the resonant tunneling scenario in the manuscript. As Reviewer #3 pointed out, the resonant tunneling scenario requires fine-tuning to explain the actual experiment, which is also noted in Ref. 28. Unless reasonable tunneling rates are theoretically predicted, the resonant tunneling scenario is just a possible scenario to explain the experimentally confirmed intrinsic nucleation.

We would like to thank the editors and all the referees for their responses.

We are gratified by the strong recommendations of referees 1 and 2 for publication in Nature Communications.

We agree that the correct balance must be struck between the experimental demonstration of an intrinsic nucleation mechanism, and its interpretation.

In the previous versions of the manuscript, it was our intention to offer the resonant tunneling model (RTM) as a hypothesis to explain the observed intrinsic nucleation. We find this scenario to be appealing because of the special features that apply under confinement, arising from the putative stripe phase (and its intrinsic tuneability via stripe periodicity). Furthermore the manuscript offers proposals to test, and perhaps refute, our bold hypothesis. Clearly more theoretical work, beyond the scope of our manuscript and outside the domain of our expertise, is highly desirable.

In view of the strong suggestion of reviewers 3 and 4 we have modified the manuscript in several ways to better acknowledge "...the preliminary nature of the resonant tunneling interpretation", in the context of other possible models

We should point out that supplement 3, devoted to a brief discussion of the RTM, was added specifically at the suggestion of one of the reviewers (Ref 1) in a previous round. We believe it should remain in the supplemental material as it adds to the accessibility of the model and to our hypothesis.

The changes made to the main manuscript are:

Page 4: Changed language introducing the RTM as a possible explanation: Here we argue that the presence of such a stripe phase **could mediate the A to B transition. A possible scenario for such a transition is the** resonant tunneling mechanism for B-phase nucleation.

Page 9: Modified the language to de-emphasize the RTM and emphasize the hypothetical nature of the intermediate states. **A scenario that is consistent with the small supercooling and draws on the putative stripe phase is** ~~On the other hand~~ the Tye-Wohns mechanism²⁸ **which (as proposed) invokes the presence of** ~~an intermediate state~~ **intermediate hypothetical states** between the A phase and the B phase which aids nucleation via resonant tunneling.

Page 10: added a phrase to the text -- the stripe phase is not stable but provides the required nearby set of quantum vacua **(for example via resonant tunneling or another competing mechanism)** to nucleate the B phase. (see Supplementary Note 3).

Page 10: Modified language To **test validate** the resonant tunneling scenario (~~model~~) one would have to quickly traverse the region. [This emphasizes that the RTM isn't established.]

Page 12: dropped the reference to the resonant tunneling model so that the last sentence of the top paragraph reads: pressure window over which a "nearby" stripe phase may play a role in B phase nucleation. ~~via the resonant tunneling mechanism.~~

Page 12-13: Modified the text to now read: **A possible mechanism for the efficient nucleation of the B phase invokes resonant tunneling** where even if not stable, the stripe phase **with its spatially modulated order parameter** can provide the intermediate metastable **states** to mediate the transition. Here we removed the words “However the observations of B phase nucleation are consistent with the resonant tunneling scenario where we propose that the stripe phase can provide the intermediate metastable states” in favor of the language above and added an explanatory phrase: the stripe phase **with its spatially modulated order parameter** can provide the intermediate metastable **states** to mediate the transition.

Page 13: **should provide a further test of the potential nucleation scenarios** instead of “a further test of our hypothesis.”

Page 13: Modified statement by adding “**or another competing mechanism**” to read: ... promote B phase nucleation, via resonant tunneling **or another competing mechanism**.

Page 21: Caption to Fig. 4 Deleted “**suggestive of the stripe phase mediated resonance tunneling mechanism of the B phase nucleation**”

Modifications were also made to the supplementary material (highlighted below) to language in the last and penultimate paragraph of supplementary note 3:

We **propose** that these virtual or real “striped phase variants” **may** function as intermediaries ... should test this nucleation **scenario**.
suppress the **putative** resonant tunneling are also envisaged

We feel that all these changes accurately reflect the status of the RTM model as a possible scenario, in a way not to mislead the objective reader.

However, we would like to comment further on the text of the response of reviewers 3 and 4. Reviewer 4 agrees with reviewer 3 that the resonant tunneling scenario requires “fine tuning”, and refers to the original paper of Tye and Wohns (ref 28). Of course that paper applies to the bulk superfluid phases. It is important to recognize that the situation under confinement differs because of the putative stripe phases: the periodicity (perhaps even morphology) of spatially modulated superfluid phases, in addition to the bowing of the cavity height leads essentially to a multiplicity of intermediate phases. This could be the essential new ingredient that “turns on” the resonant tunneling scenario, under confinement. We hope this new idea, “a possible scenario” which is certainly not a claim to definitive interpretation, will stimulate more theoretical and experimental work.

We hope that in the light of this, we believe reasonable, explanation of our approach and with the changes made to text, that the manuscript will meet with the approval of the editors and the reviewers and that we can proceed forward.